# Integrative Evaluation of the Clinical Significance Underlying Protein Arginine Methyltransferases in Hepatocellular Carcinoma

**DOI:** 10.3390/cancers15164183

**Published:** 2023-08-20

**Authors:** Yikun Jiang, Shibo Wei, Jin-Mo Koo, Hea-Ju Kim, Wonyoung Park, Yan Zhang, He Guo, Ki-Tae Ha, Chang-Myung Oh, Jong-Sun Kang, Jee-Heon Jeong, Dongryeol Ryu, Kyeong-Jin Kim, Yunju Jo

**Affiliations:** 1Department of Orthopedics, The Second Hospital of Jilin University, Changchun 130041, China; 2Department of Precision Medicine, Sungkyunkwan University (SKKU) School of Medicine, Suwon 16419, Republic of Korea; weishibo@g.skku.edu (S.W.);; 3Department of Korean Medical Science, School of Korean Medicine, Pusan National University, Yangsan 50612, Republic of Korea; 4Department of Molecular Cell Biology, Sungkyunkwan University School of Medicine, Suwon 16419, Republic of Korea; 5Department of Obstetrics and Gynecology, The Second Hospital of Jilin University, Changchun 130041, China; 6Department of Biomedical Science and Engineering, Gwangju Institute of Science and Technology, Gwangju 61005, Republic of Koreadryu@gist.ac.kr (D.R.); 7Department of Biomedical Sciences, College of Medicine, Inha University, Incheon 22212, Republic of Korea; 8Research Center for Controlling Intercellular Communication (RCIC), College of Medicine, Inha University, Incheon 22212, Republic of Korea

**Keywords:** protein arginine methyltransferase, protein methylation, hepatocellular carcinoma, T cell exhaustion, tissue microarray, prognostic marker

## Abstract

**Simple Summary:**

Protein arginine methyltransferases (PRMTs) are a group of enzymes that play pivotal roles in post-translational modifications (PTMs) and are involved in the regulation of various cellular processes. Although PRMTs are recognized for their impact on the development and prognosis of various cancers, their specific involvement in hepatocellular carcinoma (HCC) remains relatively limited. In this study, we systematically evaluated the association between PRMTs and HCC prognosis, revealing their potential as prognostic biomarkers. We also highlighted the importance of PRMT1 and elucidated its role in HCC progression, especially in the context of T cell exhaustion (Tex). Our study underscores the prospective utility of PRMTs, particularly PRMT1, as promising prognostic indicators for HCC, as well as therapeutic targets for this malignancy.

**Abstract:**

HCC is a major contributor to cancer-related mortality worldwide. Curative treatments are available for a minority of patients diagnosed at early stages; however, only a few multikinase inhibitors are available and are marginally effective in advanced cases, highlighting the need for novel therapeutic targets. One potential target is the protein arginine methyltransferase, which catalyzes various forms of arginine methylation and is often overexpressed in various cancers. However, the diverse expression patterns and clinical values of PRMTs in HCC remain unclear. In the present study, we evaluated the transcriptional expression of PRMTs in HCC cohorts using publicly available datasets. Our results revealed a significant association between PRMTs and prognosis in HCC patients with diverse clinical characteristics and backgrounds. This highlights the promising potential of PRMTs as prognostic biomarkers in patients with HCC. In particular, single-cell RNA (scRNA) sequencing analysis coupled with another human cohort study highlighted the pivotal role of PRMT1 in HCC progression, particularly in the context of Tex. Translating these findings into specific therapeutic decisions may address the unmet therapeutic needs of patients with HCC.

## 1. Introduction

HCC, a malignancy originating from the liver, accounts for a substantial number of cancer-related fatalities worldwide. Recent statistics indicate that HCC was the sixth most common cancer and the third leading cause of cancer-related deaths worldwide in 2020 [1]. Epidemiologic studies have indicated that men have a higher incidence and mortality rate than women, with rates two to three times higher in men. HCC is also commonly referred to as the “Asian cancer”, as it has the highest incidence rate in Asia-Pacific regions, such as Mongolia and Thailand [2]. HCC is a pathologically heterogeneous malignancy that accounts for approximately 90% of all primary liver cancers [3]. Treatment of HCC is primarily influenced by the tumor-node-metastasis (TNM) staging system and the functional status of the liver [4]. Although surgical interventions, such as liver resection and transplantation, are the primary curative treatment options for HCC, their applicability is limited to patients with early stage disease and preserved liver function [5]. Unfortunately, only a small proportion of patients with HCC (estimated at 30%) are eligible for surgical intervention due to the advanced stages of the disease or underlying liver cirrhosis. In addition to surgery, there are several alternative therapeutic modalities for early stage HCC. These include radiofrequency ablation, transarterial chemoembolization, and pharmacological agents such as multikinase inhibitors including sorafenib and lenvatinib [6]. However, as HCC progresses to advanced stages, curative interventions may no longer be feasible, and supportive care remains the only feasible option, which, taken together, result in a high mortality rate of HCC [7]. This underscores the significance of timely diagnosis and accurate management of HCC to enable early initiation of treatment and maximize the likelihood of therapeutic success. Meanwhile, the scarcity of curative therapies and the resulting poor prognosis make elucidating the biology of HCC and translating it into viable treatments a priority [8].

PRMTs are a class of enzymes involved in the PTMs of proteins, which catalyze the methylation of arginine residues [9]. PRMTs play key roles in various biological processes, such as RNA metabolism, signal transduction, gene expression, epigenetic regulation, autophagy, and cardiac function [10,11,12,13]. The PRMT family comprises nine members [14]. Recent studies have highlighted the significance of PRMTs in the development of multiple human disorders, including cardiovascular diseases, neurological disorders, viral infections, and cancers [15,16]. Methylation of numerous protein substrates catalyzed by PRMTs play a pivotal role in various cancer processes, including initiation, progression, and metastasis. Dysregulation of PRMT1 expression has been consistently identified in diverse human malignancies, including breast, lung, colon, and bladder cancers, as well as leukemia [9], with extensive research elucidating its regulatory role and substantial prognostic significance in these cancers [17,18]. PRMT2 is upregulated in breast cancer and glioblastoma progression [19], and is positively correlated with poor clinicopathological characteristics and survival outcomes in renal cancer [20]. A similar pattern was observed for PRMT3 in colorectal cancer [21]. PRMT4 has been experimentally proven to promote cancer cell migration and metastasis in high-grade breast tumors and lymphomas and serves as a biomarker for well-differentiated breast cancer [22]. PRMT5 overexpression is a recognized prognostic indicator of poor survival in several cancers [23,24]. Consistently, PRMT6 expression is associated with growth promotion in lung tumors [25,26]. PRMT7 overexpression has been linked to renal cancer and is correlated with a poor prognosis [27]. PRMT8 expression has been reported to have conflicting prognostic implications in different cancer types, with elevated levels being associated with improved patient survival in breast and ovarian cancer, but poorer survival in gastric cancer [28]. 

However, the current research on the role of PRMTs in HCC remains relatively limited. Previous studies have suggested a potential association between PRMTs and HCC progression, which is largely manifested by the dysregulated expression of certain PRMTs during HCC development. For example, elevated levels of PRMT1 and PRMT2 have been detected in HCC patients, underscoring their potential implications in disease pathogenesis [29,30]. Conversely, diminished expression of PRMT4 in malignant liver cancers suggests its plausible role as a tumor suppressor [31]. Another report suggested the opposite, implicating PRMT4 in an autophagy pathway that is important for liver carcinogenesis [32]. Furthermore, overexpression of PRMT9 has been linked to poor outcomes in HCC, demonstrating an augmenting effect on the metastatic potential of cancer cells [33]. Additionally, the abnormal expression of typical PRMTs has been associated with diverse clinicopathological characteristics in patients with HCC, including microvascular invasion, poor tumor differentiation, larger tumor size, presence of portal vein tumor thrombus, and elevated alpha-fetoprotein levels [29]. Drawing upon the existing evidence, it is plausible to postulate that PRMTs may play a multifaceted role in the pathogenesis of HCC and serve as indicators of disease progression or prognosis. Nonetheless, owing to the lack of comprehensive understanding of PRMTs in HCC, we aimed to conduct a systematic evaluation of their expression patterns and prognostic implications within the intricate landscape of HCC. Our findings shed light on the monitoring of HCC progression and identifying avenues for developing effective therapies against HCC.

Of note, accumulating evidence reveals that HCC triggers an inflammatory reaction, resulting in immunotherapy, in which the immune response is primed to strike malignancy, as a viable option [34]. However, despite the efficacy in experiments and the limited benefits to patients from immunotherapy, most clinical trials have been disappointing, with no signs of clear curative effects or prognosis amelioration. Under these circumstances, there is an urgent need to understand the functional status of immune cells and the immune environment in contexts ranging from molecular intricacies to HCC development. Tumor-specific effector CD8^+^ T lymphocytes, which enable the distinction of relevant tumor antigens within autologous tumor tissues, have been established to be intimately involved in HCC management, not only for tumor growth suppression but also for prolonged progression-free survival, either driven by anti-tumor treatment or occurring naturally [35]. However, under repressive tumor microenvironments and extended antigen exposure, CD8^+^ T cells tend to progress to a phage termed Tex. Tex encompasses the malfunctioning stages that occur within antigen-specific CD8^+^ T lymphocytes. It was initially characterized in the context of chronic viral infection and spanned into most tumors, with the expression of multiple inhibitory receptors, such as PD-1, implicated in immune checkpoint blockade and immune escape. Recent studies have revealed that Tex plays a crucial role in HCC progression; however, its specific molecular mechanisms remain largely undefined. Notably, certain PRMTs have been shown to be linked to T cell regulation. For instance, PRMT1 plays a critical role in modulating T cells by regulating memory T cell polyfunctionality [36] and T cell activation [37]. But hitherto, there is a lack of research highlighting the putative relevance of PRMTs on Tex in HCC progression. Therefore, we aimed to perform further evaluations to explore the underlying impact of PRMTs on Tex in the context of HCC. 

In this study, we conducted a comprehensive transcriptome analysis using three independent cohorts to elucidate the prognostic implications of PRMTs in HCC. Additionally, we employed a publicly accessible dataset to assess PRMT expression at the single-cell transcriptome level, revealing its potential involvement in the regulatory mechanisms underlying HCC development. Subsequently, we further evaluated PRMT1 expression patterns in 61 normal liver and 54 HCC samples obtained from human subjects. In conclusion, our findings indicate that PRMTs are predicted to be novel prognostic markers for HCC, and PRMT1 may perform significant effects on HCC development via Tex regulation, suggesting that targeting PRMT1 and PRMT1-positive Tex may be effective HCC therapies. 

## 2. Methods

### 2.1. Data Collection and Preprocessing 

The hepatic transcriptomes of normal liver and HCC tissues were obtained from the National Center for Biotechnology Information (NCBI) Gene Expression Omnibus (GEO) database (GSE14520 [38], GSE25097 [39], and GSE36376 [40]). The RNA-seq data and clinical information of the liver hepatocellular carcinoma (LIHC) cohort were downloaded from The Cancer Genome Atlas (TCGA) data portal (Accessed on 1 December 2020; http://portal.gdc.cancer.gov/). Normalization of HCC transcriptome data from TCGA was performed using the Fragments Per Kilobase of exon per million (FPKM) method. 

### 2.2. Gene Ontology (GO) Enrichment Analysis 

GO enrichment analysis was performed to identify the potential mechanisms underlying PRMT expression in the pathogenesis and prognosis of HCC. First, from TCGA LIHC dataset (*n* = 368), we prepared the entire transcriptome of 20 subjects showing either the highest or lowest expression of each PRMT family member. Absolute fold changes (F.C. = Abs [mean expression of highest/mean lowest]) were computed to identify the top correlated GO biological processes and human phenotype ontology (HPO). Gene sets were considered significantly enriched when the nominal *p*-value < 0.05 and the false discovery rate < 0.05. 

### 2.3. Trajectory Analysis and Gene Expression Assessment

The data utilized in this study were sourced from the publicly available NCBI GEO database, especially the accession number GSE98638 [41]. The dataset consisted of deep scRNA-seq data derived from the peripheral blood, tumors, and adjacent normal tissues of six patients diagnosed with HCC. Our analysis focused specifically on CD8^+^ T cells isolated from both the peripheral blood and tumor tissues. To explore the expression patterns of the genes of interest, we conducted a trajectory assay on the scRNA-seq data using the phate R package [42]. This computational analysis enabled us to identify specific cell types in which the target genes exhibited distinct expression levels.

### 2.4. Clinical Specimens for Prognostic Relevance Evaluation via Immunofluorescent Staining

Liver tissues (cancer and para-cancer) were collected from patients with HCC and settled as tissue microarray by the Shanghai Outdo Biotech Company (CGT NO.: HLivH180Su11; Lot No.: XT16-029), with detailed clinical information and prognosis of patients. Immunofluorescent staining was performed using the Opal 7-color Manual IHC Kit (NEL801001KT, PerkinElmer). DAPI staining was used to stain the nuclei. After staining, the tissue microarray was panoramically multispectrally scanned using the TissueFAXS system, and the data were analyzed using StrataQuest software. A single-channel fluorescence signal was obtained after spectral splitting using a spectral database. Effective nuclei were identified and screened according to the DAPI channel, which was identified as the core for determining the distance radius under the staining conditions of each protein channel. Fluorescence signals were subsequently measured. The threshold for each channel was set separately based on the staining status. The positive cell population was then divided to measure the positivity rates and fluorescence intensity of each target in different cells. 

### 2.5. Data Analysis, Visualization, and Statistics

All bioinformatics data were analyzed and visualized as previously described [43]. All analyses and visualizations were performed using R (version 4.1.0; https://www.r-project.org), Rstudio (version 2023.03.0; https://www.rstudio.com/), and R packages, as listed below:Data preprocessing: dplyr, stringr, reshape2, naniar, and skimr.Boxplot and survival curve: ggplot2, ggpubr, pROC, multipleROC, survival, survminer, egg, gridExtra, and grid.

The normality of all data was determined using the Shapiro–Wilk test. Differences between two and three groups were evaluated using the Wilcoxon rank-sum test or the Kruskal–Wallis test, respectively. To evaluate survival probability, we first used the R package *multipleROC* to determine the optimal gene expression value. These values were used to divide the samples into two groups. Kaplan–Meier curves were then generated with the R packages, *survminer* and *survival*. 

## 3. Results

### 3.1. Expression Patterns of PRMTs in Human HCC

To confirm the association of PRMTs with HCC and evaluate the diagnostic potential of HCC, we conducted an investigation using three independent HCC cohorts from the GEO database. In GSE14520, the expressions of *PRMT1*, *PRMT3*, *PRMT4*, and *PRMT5* were upregulated in cancer tissues (Figure 1A), whereas *PRMT2* and *PRMT8* were downregulated. There were no significant differences in *PRMT7* expression level. Data on the expressions of *PRMT6* and *PRMT9* have not been validated. In cancer tissues of GSE25097, the expressions of *PRMT2*, *PRMT3*, *PRMT4*, *PRMT5*, and *PRMT7* were upregulated in cancer tissues (Figure 1B), whereas the expressions of *PRMT6* and *PRMT9* were downregulated. There were no significant differences in the expressions of *PRMT1* and *PRMT8*. In GSE36376, the expressions of *PRMT1*, *PRMT2*, *PRMT3*, *PRMT4*, *PRMT5*, and *PRMT7* were upregulated in cancer tissues (Figure 1C). There were no significant differences in the expressions of *PRMT6*, *PRMT8*, and *PRMT9*. To further estimate the potential of PRMTs as diagnostic markers for HCC, we conducted an Area Under the Receiver Operating Characteristic curve (AUROC) analysis on those three independent datasets (Figure 1D). The AUROC scores of PRMTs also indicated that *PRMT1*, *PRMT3*, *PRMT4*, and *PRMT5* were associated with HCC, demonstrating their diagnostic potential. In summary, although the expression patterns of some PRMTs could not be concluded, *PRMT1*, *PRMT3*, *PRMT4*, and *PRMT5* are likely to be stronger diagnostic biomarkers with a substantial correlation to HCC tumorigenesis. Others, such as *PRMT2* and *PRMT8*, exhibit a lack of unanimity in terms of ambiguous, even contradictory, expression patterns that necessitate further evaluation based on their clinical contexts.

### 3.2. PRMT Expression Is Associated with Clinicopathological Characteristics in Human HCC

To discern the prognostic potential of PRMTs in HCC development, we systematically evaluated the correlation between PRMT expression and clinicopathological features in The Cancer Genome Atlas (TCGA) patients with liver hepatocellular carcinoma (LIHC). First, we classified and scrutinized the mRNA expression levels of PRMTs in 344 patients with HCC according to their TNM stages (Figure 2). Owing to the inherent scarcity of patients with HCC, especially those in TNM stages III and IV, we grouped patients from both stages III and IV to facilitate robust statistical analysis and reduce the risk of bias. In the cohort, 170 patients were categorized as stage I, 86 patients were categorized as stage II, and 88 patients were categorized as stages III + IV. Notably, the expressions of *PRMT1* and *PRMT2* demonstrated increases with the advancement of clinicopathological conditions. Likewise, HCC patients with stages III + IV showed a significant increase in the expression levels of *PRMT3*, *PRMT4*, and *PRMT5* compared to those with stage I. Conversely, no notable differences were observed in the expression levels of *PRMT6*, *PRMT7*, and *PRMT9*. Collectively, our findings suggest that upregulated PRMTs have a possible involvement in the progression of HCC. Notably, our study highlights the potential of *PRMT1* and *PRMT2* as promising prognostic biomarkers for HCC, given their correlation with advanced TNM stages.

### 3.3. Prognostic Significance of PRMT Expression in Human HCC

To evaluate the prognostic potential of PRMTs in HCC patients, we performed an assessment using Kaplan–Meier estimation. Our analysis revealed a correlation between each PRMT and the survival outcomes of 344 HCC patients. We classified patients into high- and low-expression groups based on the optimal expression levels of PRMTs, as determined by multiple receiver operating characteristic curve analyses [44]. Strikingly, our investigation of the overall survival of HCC patients revealed a significant correlation between PRMTs and poor prognosis in HCC (Figure 3). Specifically, overexpression of *PRMT1*, *PRMT2*, *PRMT3*, *PRMT4*, *PRMT5*, *PRMT6*, and *PRMT7* were associated with reduced overall survival rates, with *PRMT1*, *PRMT3*, and *PRMT5* exhibiting the most pronounced effects. Conversely, *PRMT9* displayed an inverse association, whereby patients with lower *PRMT9* expression levels had a worse prognosis. Importantly, these findings were consistent with the expression patterns of the majority of PRMTs in HCC, as compared to normal liver tissue and TNM stages. 

To further investigate prognostic significance, we evaluated the survival rates in HCC patients at different TNM stages. To facilitate comparative analyses between early and advanced HCC, as well as to address the potential limitations of sample size in each stage, patients were divided into early HCC with TNM stages I + II (256 patients) and advanced HCC with TNM stages III + IV (88 patients), respectively. In patients with TNM stages I + II HCC, high expression of *PRMT1*, *PRMT3*, and *PRMT6*, and low expression of *PRMT9* were significantly associated with decreased survival (Figure 4A), with *PRMT1* displaying the most pronounced effect. However, no significant correlation was observed between the expression levels of *PRMT2*, *PRMT4*, *PRMT5*, and *PRMT7* and the survival rate, although *PRMT2* and *PRMT7* displayed a trend similar to that of the overall survival rates. Herein, we also estimated the survival probability of HCC patients at TNM stages I and II (Appendix A). In stage I, high expression of *PRMT1* and *PRMT7*, and low expression of *PRMT9*, showed statistical correlation with declined survival outcomes. In stage II, high expression levels of *PRMT1*, *PRMT3*, and *PRMT6* showed statistical correlation with adverse survival outcomes. Considering the apparent consistency between separated and merged stages I and II, we opted for a grouping strategy to compare HCC progression while addressing the limitations imposed by the sample size. Although the cohort of TNM stages III + IV HCC was insufficient to draw definitive conclusions, the current results demonstrated a certain association between high expression of *PRMT1*, *PRMT3*, *PRMT4*, *PRMT5*, and *PRMT6* and poor survival rates (Figure 4B), with *PRMT1* and *PRMT5* showing notable effects, while no significant correlation was observed for other PRMTs. Collectively, these trends in the survival rates for each TNM stages were broadly consistent with those observed in the overall analysis.

### 3.4. Gender- and Ethnicity-Specific Correlations of PRMT Expression in Human HCC

To investigate whether PRMT expression has gender-specific differences in HCC, we conducted a survival analysis of 110 female and 234 male patients. In females, we observed a significant association between high *PRMT4* and *PRMT6* and poor survival rates (Figure 5A). However, there was no significant correlation between the expression levels of *PRMT1*, *PRMT2*, *PRMT3*, *PRMT5*, and *PRMT9* and the survival rate. Notably, *PRMT7* displayed a completely different trend, resulting in a poor survival rate at higher expression levels, albeit with the possibility of data size limitation artifacts. In males, we observed a significant association between high expression of *PRMT1*, *PRMT2*, *PRMT3*, *PRMT5*, *PRMT7*, and low expression of *PRMT9* and poor survival rates (Figure 5B). Conversely, no significant correlation was observed between the expression levels of *PRMT4* and *PRMT6* and survival rate. These results suggest the possibility of gender-specific effects of PRMT expression on HCC development, which warrant further investigation in larger patient cohorts and in-depth mechanistic studies.

Given the higher incidence of HCC [2] and the unveiled correlation between certain PRMTs and HCC risk factors [45,46] in Asian countries, we also estimated the survival rates of a cohort of 156 Asian and 164 Caucasian patients to determine the potential correlation between PRMT expression and HCC outcomes in different races using hazard ratio analysis. Our findings revealed a significant association between high expression levels of *PRMT1*, *PRMT2*, *PRMT3*, *PRMT4*, *PRMT5*, *PRMT6*, and *PRMT9* and poor survival rates in the Asian cohort (Figure 6A). In contrast, among Caucasians, the association between PRMT expression and HCC survival was race-specific. High expression of *PRMT3*, *PRMT4*, *PRMT5*, and *PRMT6*, and inversely low expression of *PRMT2*, were associated with poor survival (Figure 6B). These findings suggest that the underlying biological differences between races influence the impact of PRMT expression on HCC development and prognosis.

### 3.5. Single-Cell Transcriptomic Analysis of CD8^+^ T Cell Suggests an Association between PRMT1 Expression and Tex 

To elucidate the mechanisms underlying the association between PRMT expression and HCC prognosis, we conducted a comprehensive analysis of PRMT expression at the single-cell transcriptome level using a publicly available dataset [41]. Specifically, we focused on CD8^+^ cells derived from HCC patients and assessed their association with immune checkpoints. Our analysis revealed the presence of CD8^+^ cells in both peripheral blood and within the HCC tumor microenvironment (Figure 7A). Interestingly, the expression of *PRMT3-9* did not display clear correlations with specific cell subpopulations and was relatively low, whereas *PRMT1* and *PRMT2* exhibited relatively high expression or were specifically expressed in certain subpopulations (Figure 7B). Notably, *PRMT1* was found to be expressed in a subset of CD8^+^ T cells derived from HCC. We further investigated the relationship between *PRMT1* expression and immune checkpoints, which are known to play a crucial role in Tex (Figure 7C,D). Strikingly, our analysis revealed a concurrence of *PRMT1* expression with exhausted T cell markers, including *CTLA4*, *CXCL13*, *ENTPD1*, *HAVCR2*, *LAG3*, *SIRPG*, and *PDCD1* (Figure 7D). This observation provides a potential explanation for the poor prognosis associated with high *PRMT1* expression. Based on existing analyses, we hypothesize that PRMT1 expression may be associated with Tex in HCC.

### 3.6. Verification of PRMT1 Expression Patterns in Human HCC and Its Potential Role in Tex

To verify our hypotheses generated from the LIHC database analysis and corroborate the association of PRMT1 with HCC that was identified through our single-cell transcriptome analysis, we recruited Asian patients with HCC and procured a cohort of 86 healthy liver tissues and 92 HCC specimens. Herein, we focused on the prognostic significance of PRMT1 in HCC and elucidated its underlying association with Tex. In this case, the collected human samples were processed into microarray, followed by immunofluorescent staining. After removing samples with poor quality staining, including those that were negative for PD-1 or CD8^+^, 61 normal liver tissues and 54 HCC samples were included in subsequent analysis. As a proof-of-concept, HCC samples demonstrated a larger proportion of PD-1^+^ and a lower proportion of CD8^+^ compared to normal tissues (Appendix A), conformant to the fluorescent intensity of PD-1 and CD8^+^ staining, respectively (Appendix A). Representative co-staining images showed PRMT1 overexpression in HCC samples (Figure 8A), corroborated by a higher percentage of PRMT1 positive (PRMT1^+^) cells, as well as higher intensity of PRMT1 (Figure 8B). Intriguingly, the percentages of PRMT1-PD-1^+^-CD8^+^ positive (PRMT1-Tex^+^) cells and PRMT1 expression intensity in PRMT1-Tex^+^ cells were higher in HCC samples in comparison to normal tissues (Figure 8C). However, despite the clear trend towards PRMT1 overexpression in PRMT1-Tex^+^ cells, there were no significant differences between the two groups, primarily due to the dispersion of the values. As such, we partitioned the data into two groups, high expression (top 25%) and low expression (bottom 75%), for both PRMT1^+^ and PRMT1-Tex^+^. Subsequent comparisons of PRMT1^+^ high (PRMT1^Hi^) and PRMT1^+^ low (PRMT1^Lo^) expression levels between normal and HCC samples reinforced our initial assumption that PRMT1 displayed statistically higher expression in terms of both PRMT1^Hi^ and PRMT1^Lo^ groups (Appendix A), which were of comparable results to the PRMT1^Hi^-Tex^+^ and PRMT1^Lo^-Tex^+^ groups (Appendix A). To further verify the association between PRMT1 expression and clinicopathological outcomes in HCC patients, we categorized PRMT1 expression in HCC and PRMT1 expression in Tex of HCC into two groups, as previously outlined (Figure 8D,E), with no statistical difference in clinical characteristics (Appendix A). Strikingly, the expression patterns of PRMT1^Hi^, including the positivity rate (Figure 8F) and expression intensity (Appendix A), in both HCC and Tex were closely associated with a poor prognosis in HCC, which was consistent with our bioinformatics analysis. Comparable results were concluded from the similar expression patterns in male patients (Figure 8G) and at each TNF stage (Figure 8H,I), underscoring the relevance of PRMT1 in the poor prognosis of HCC and its potential impact on Tex in HCC, especially in male patients.

## 4. Discussion

It is well known that HCC is a prevalent primary liver cancer that has a significant impact on global health [47]. However, the optimal therapeutic approach for HCC remains a subject of ongoing debate due to the paucity of consensus regarding the molecular targets and underlying mechanisms driving HCC progression [48]. To improve the clinical management of HCC, it is crucial to diagnose the disease at an early stage and tailor treatments based on molecular targets. The present study aimed to address this need by conducting a comprehensive analysis of the gene expression patterns of the PRMT family to identify the key members that play an essential role in HCC gene therapy.

PRMTs are a family of enzymes that facilitate the transfer of methyl groups from S-adenosylmethionine to arginine residues, resulting in mono- or di-methylated arginine residues. These methylated arginine residues are involved in indispensable PTMs that regulate a wide range of cellular processes, such as transcription, splicing, and DNA damage response. Through this regulatory function, PRMTs modulate protein functions and interactions, ultimately affecting cellular homeostasis and disease states [49]. Although a survey within the Ensembl database predicted the existence of all PRMT family members, only the expression of PRMT1, 2, 4, and 7 has been characterized and confirmed in mammalian cells [50,51,52]. Interestingly, most of these members were found in cancer cells, suggesting their potential involvement in cancer biology. However, the precise role of PRMTs in cancer remains elusive and requires further investigation. Our study addresses this gap by providing evidence that PRMTs play crucial roles in the development of HCC and have the potential to serve as prognostic biomarkers and, by extension, as effective therapeutic targets for HCC.

In our study, we observed a consistent upward trend of PRMTs, particularly PRMT1, 3, 4, and 5, in HCC tissues compared to normal liver tissues, by horizontal comparison of PRMT expression patterns across three independent HCC cohorts, as well as by AUROC analysis. Our findings indicate that certain PRMTs have great potential as diagnostic markers for HCC tumorigenesis. To further assess the prognostic value of PRMTs in HCC progression, we conducted a systematic evaluation using the TGCA LIHC dataset. Analysis of variations in PRMT expression in patients with diverse clinicopathological features implied a strong positive correlation between PRMT1 and 4, with a pronounced emphasis on PRMT1 and HCC progression. These findings underscore the potential of PRMT1 as an indicator of pathological advancement in HCC. To further investigate the relationship between PRMT expression and survival outcomes in HCC, we performed Kaplan–Meier analysis. Our analysis revealed a positive association between the overall expression of PRMTs and poor prognosis in HCC, with the exception of PRMT9, which showed an inverse correlation. Upon grouping the patients according to TNM stage for further examination, we observed that certain PRMTs, including PRMT4 and 5, were exclusively correlated with the prognosis of HCC in advanced stages. In contrast, PRMT1, 3, and 6 exhibited prognostic significance throughout the entire pathological progression of HCC, with PRMT1 displaying the most significant difference. This suggests that PRMT4 and 5 could be employed as efficient indicators for prompt prognosis in advanced patients, assisting in the evaluation of clinical processes. Notably, PRMT2 displayed contradictory expression patterns, with a marginal decline observed in stage III/V compared to that in stage II, whereas elevated PRMT2 expression was negatively correlated with the overall survival outcomes of HCC patients. A recent study revealed that PRMT2 could induce HCC tumorigenesis through Bcl2 activation [30], which substantiates the notion that PRMT2 plays an influential role in driving HCC development, as evidenced by the significant upregulation observed in stages II and III/V compared with stage I. In this case, it is plausible to postulate that PRMT2′s impact might be predominantly confined to the early stages of HCC tumorigenesis, with limited or even counteractive influences on the later stages regulated by complex mechanisms. Nevertheless, there is a lack of comprehensive investigations in this domain, underscoring the need to delve deeper into the distinct mechanistic roles of PRMT2 in HCC progression. Additionally, we observed an intriguing contrast regarding PRMT6, as it appeared to be irrelevant to TNM stage but demonstrated prognostic significance for survival outcomes based on our Kaplan–Meier analysis. While the lack of further clarification leaves the underlying explanation unclear, there is a possibility of obtaining certain clues from our finding of a PRMT6 decline in HCC compared to normal tissues, along with a recent study indicating that PRMT6 deficiency promotes autophagy induction in HCC [53]. The conflicting impact of PRMT6 might be attributed to the regulation of autophagy processes triggered by the progressive decline in PRMT6 expression during HCC tumorigenesis and progression, thereby resulting in bidirectional effects, including both tumor development and suppression [54]. However, to fully elucidate the specific underlying mechanisms, further studies are imperative. Conversely, the consistent prognostic significance of PRMT1 throughout the entire HCC course suggests its role in the general pathological progression of HCC and its potential impact on patient outcomes. This finding highlights PRMT1′s potential as a valuable prognostic marker for the duration of HCC and provides clinicians with crucial information for patient management and treatment decisions. Thus, our study provides compelling evidence for the correlation between PRMT expression patterns and HCC progression. Specifically, PRMT1 stands out as a key player in the pathological advancement of HCC, making it a promising indicator of disease severity and patient prognosis.

Additionally, we made interesting observations regarding the gender specificity of HCC, noting its higher prevalence in men compared to women, which is consistent with previous research findings [1]. This gender specificity may be attributed to various factors, including hormone levels, genetic predisposition, and lifestyle [55]. Nevertheless, the molecular mechanisms underlying this phenomenon may affect the clinical efficacy and prognosis of HCC patients. To elucidate this, we investigated gender-specific correlations with PRMT expression in HCC. Our findings revealed that PRMT1 was closely associated with male prognosis, whereas PRMT4 and 6 were linked with female prognosis. Intriguingly, we observed an inverse correlation between PRMT7 and gender. These results imply that PRMTs are widely involved in regulating gender differences in the occurrence and prognosis of HCC patients, providing valuable insights for future studies on the mechanisms of HCC and paving the way for the development of precision medicine for clinical patients. Furthermore, recent research has demonstrated a marked incidence of and mortality from HCC across multiple Asian countries such as Mongolia, Thailand, Cambodia, and Vietnam [2]. This phenomenon has been postulated to be multifactorial in nature, with a greater prevalence of chronic infections caused by hepatitis B and C viruses in Asian populations, along with cultural dietary practices involving high alcohol consumption, smoking, and other unidentified environmental factors [56]. Apart from these, genetic factors may also represent another primary etiological driver of HCC, as certain genetic variations have been associated with a heightened risk of developing this malignancy. Recent research has highlighted the regulation impact of PRMTs on HCC risk factors, notably liver fibrosis, which is correlated with different lifestyles among races. Hepatic PRMT1 regulates the protein nitrosylation and participates in mediating the progression of alcohol-induced liver fibrosis [45]. Additionally, alcohol-mediated loss of PRMT6 contributes to alcohol-induced fibrosis by reducing integrin methylation and increasing profibrotic signaling [46]. These findings suggest that diverse lifestyle factors may serve as crucial determinants of racial disparities in HCC incidence and that PRMTs play an extensive role in this process. In light of this, we investigated the correlation between PRMT expression and clinical outcomes in HCC patients from diverse racial backgrounds, including both Asian and Caucasian populations. Our results revealed a noteworthy association between overexpression of PRMTs, particularly PRMT1, 2, 3, and 5, and poor prognosis in HCC patients of Asian descent rather than in Caucasians. These findings suggest that the higher incidence of HCC in Asian countries may be partly attributed to the underlying regulation of PRMTs. However, further studies are needed to authenticate this presumption and to explore the molecular mechanisms underlying the differences in PRMT expression and its impact on HCC outcomes. Collectively, our study provides valuable insights into the gender-specific and racial disparities in HCC and their potential association with PRMT expression. Understanding the molecular basis of these correlations may open new avenues for personalized approaches for HCC treatment, ultimately contributing to improved patient outcomes.

Among the comprehensive assessments employing diverse criteria for all variables from the LIHC dataset, PRMT1 has emerged as a promising prognostic biomarker, exhibiting consistent correlations with HCC. To explore the mechanisms involved, we further examined the PRMT expression profiles in CD8^+^ cells derived from patients with HCC and assessed their correlation with immune checkpoints at the single-cell transcriptome level. Intriguingly, our findings revealed the expression of *PRMT1* in a specific subset of CD8^+^ T cells that exhibited concurrent expression of markers associated with Tex. This observation provides a compelling rationale for the poor prognosis associated with elevated *PRMT1* expression in HCC. PRMT1 has been found to be clearly associated with poor prognosis in HCC patient cohorts, which is supported by our findings as well as a recent RNA-seq analysis [57], showing conformance to the latest findings on the potential promotion of PRMT1 in HCC formation by means of CDKN1A downregulation, STAT3 activation [29], and immune checkpoint PD-L1 gene overexpression [58]. Given the significance of PRMT1 in HCC pathogenesis, elucidating its mechanisms is imperative to advancing our understanding of the underlying mechanisms driving HCC and to identifying effective therapeutic targets. To this end, we conducted a further human cohort study using normal liver tissues and HCC samples from recruited Asian patients, so as to verify our hypothesis of PRMT1 as the prognostic biomarker of HCC and clarify the underlying mechanisms between PRMT1 and HCC progression. Immunofluorescent staining was conducted on microarray consisting of 61 normal liver tissues and 54 HCC samples with typical pathological features of HCC, such as overexpressed PD-1 and poorly infiltrated CD8^+^ T cells [59], the overlap of which revealed Tex, the malfunction status of T cells. In line with our previous findings, we confirmed high expression levels of PRMT1 in HCC patients, both in terms of range and intensity. Additionally, patients with higher PRMT1 expression are more prone to poor prognosis, underscoring the significance of PRMT1 as a prognostic biomarker of HCC. However, owing to limited female enrollment, we were unable to verify gender differences, which requires further clarification.

Of note, Tex is characterized by reduced cytotoxicity, decreased pro-inflammatory cytokine production, and upregulation of inhibitory receptors, which are coupled by transcriptional and epigenetic modifications [60]. These alterations contribute to immune evasion and the inhibition of inflammation during HCC progression [41]. Although numerous studies have been devoted to the decisive effects of Tex on HCC, and the accumulation of Tex in HCC is observed with worse outcomes clinically, in each case, the detailed molecular mechanisms remain unclear and of great demand to be unmasked. In our study, we made a significant observation by identifying a higher percentage of PRMT1-Tex^+^ cells in patients with HCC who overexpressed PRMT1, aligning with our scRNA-seq analysis. These findings suggest a potential correlation between PRMT1 and the presence of Tex in HCC. PRMT1-Tex^+^ has a strong positive correlation with mortality rates among diverse clinicopathological characteristics as well as in male patients. Meanwhile, HCC patients with stronger PRMT1 expression in Tex suffer worse prognoses, suggesting that PRMT1 might serve as an indicator of Tex status. Moreover, given that PRMT1 deletion leads to a reduction in PD-L1 expression, it is plausible to speculate that PRMT1 may facilitate HCC progression by boosting immune checkpoint evasion while triggering Tex. These assumptions require further evidence while providing novel perspectives for future studies on the intricate interplay between PRMT1, immune checkpoints, and Tex cells in the context of HCC.

In conclusion, we conducted a comprehensive evaluation of the expression patterns of PRMTs in HCC and established their potential value in disease development and prognosis prediction. Our findings also highlighted the significance of PRMT1 and revealed its underlying involvement in HCC progression, particularly in the context of Tex. Further research is warranted to elucidate the precise mechanisms by which PRMTs contribute to HCC and to explore the potential of targeting specific PRMTs as effective therapeutic strategies for this malignancy.

## Figures and Tables

**Figure 1 cancers-15-04183-f001:**
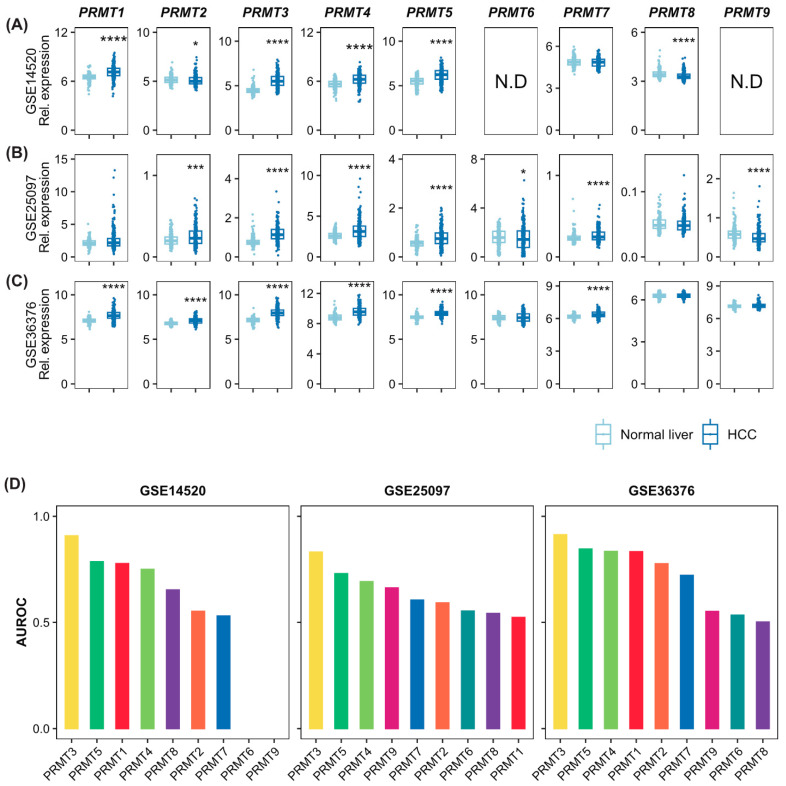
Expression levels of the protein arginine methyltransferase (PRMT) family members in normal liver and hepatocellular carcinoma (HCC) cells indicating the diagnostic potential of PRMTs in HCC. (**A**–**C**) Box and whisker plots showing the expression levels of all detected PRMTs in normal and HCC samples of three independent cohorts ((**A**). GSE14520, normal = 241, HCC = 247; (**B**). GSE25097, normal = 249, HCC = 268; (**C**). GSE36376, normal = 193, HCC = 240). The horizontal line crossing each box designates the median, while the top and bottom edges represent the first (25%) and third quartiles (75%), respectively. The dots indicate the expression value of each subject. Shapiro–Wilk test to determine the normality and Wilcoxon Rank-Sum test to compare the two groups (normal liver vs. HCC) were performed. * *p* < 0.05, *** *p* < 0.001, **** *p* < 0.0001. (**D**) The values of Area Under the Receiver Operating Characteristic Curve (AUROC) scores indicate diagnostic potential of PRMTs, implying their association with HCC.

**Figure 2 cancers-15-04183-f002:**
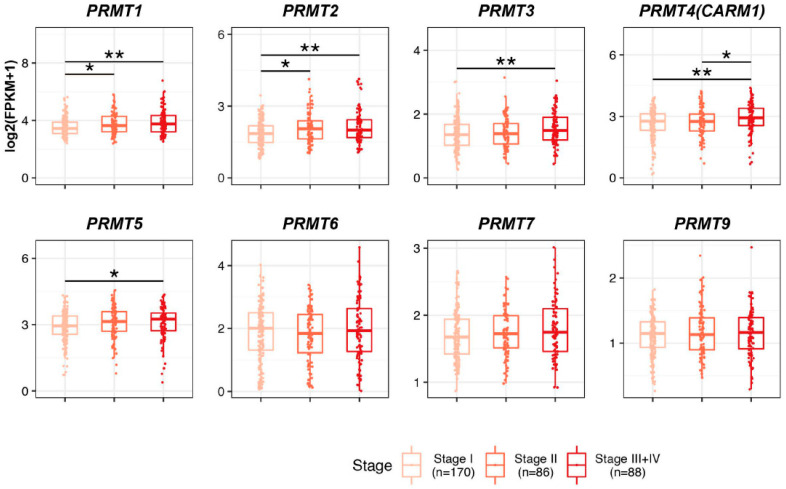
Relationship between the expression levels of PRMTs and the clinicopathological features in The Cancer Genome Atlas (TCGA) patients with liver hepatocellular carcinoma (LIHC). Boxplot showing the expression levels of PRMTs in each stage (Stage I = 170, II = 86, and III + IV = 88) in TCGA LIHC, representing the prognostic potential of PRMTs in HCC. The horizontal line crossing each box designates the median, while the top and bottom edges represent the first (25%) and third quartiles (75%), respectively. The grey dots indicate the expression value of each subject. Shapiro–Wilk test to determine the normality and Kruskal–Wallis test to compare the three groups (Stage I, II and III+IV) were performed. * *p* < 0.05, ** *p* < 0.01.

**Figure 3 cancers-15-04183-f003:**
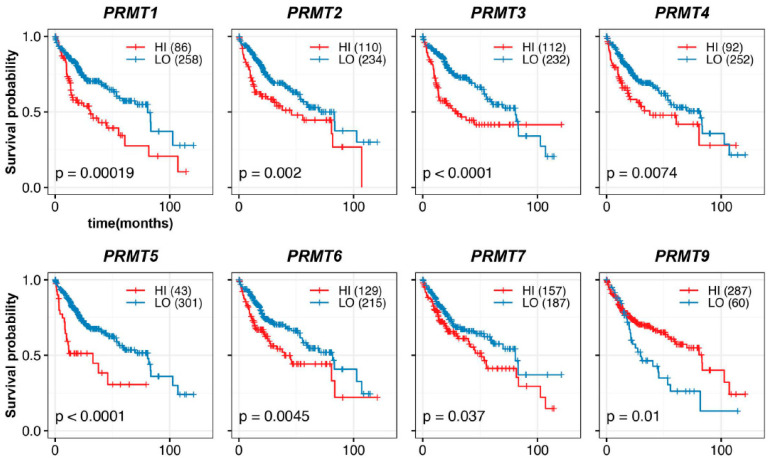
Correlation between PRMT expression and overall outcomes in HCC patients. Kaplan–Meier curves show the correlation between hepatic PRMT expression and overall survival. The R package, *multipleROC*, divided into two groups (i.e., high vs. low PRMT expression) according to the optimal gene expression level of each PRMT. The R packages, *survminer* and *survival*, estimated the survival probability.

**Figure 4 cancers-15-04183-f004:**
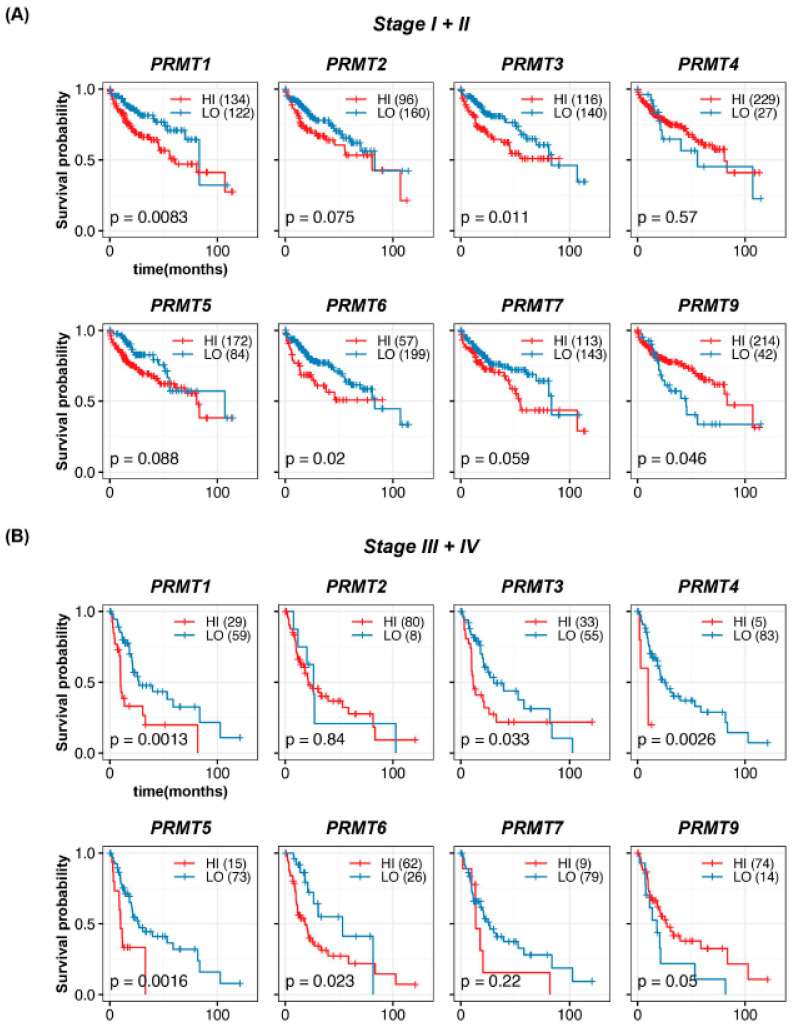
Prognostic significance of PRMTs in patients with different stages of HCC. (**A**,**B**) Survival probabilities (Kaplan–Meier curves) in different stages (Stages I + II and III + IV) of HCC show the association between hepatic PRMT expression and overall survival. The R package, *multipleROC*, divided into two groups (i.e., high vs. low PRMT expression) according to the optimal gene expression level of each PRMT. The R packages, *survminer* and *survival*, estimated the survival probability.

**Figure 5 cancers-15-04183-f005:**
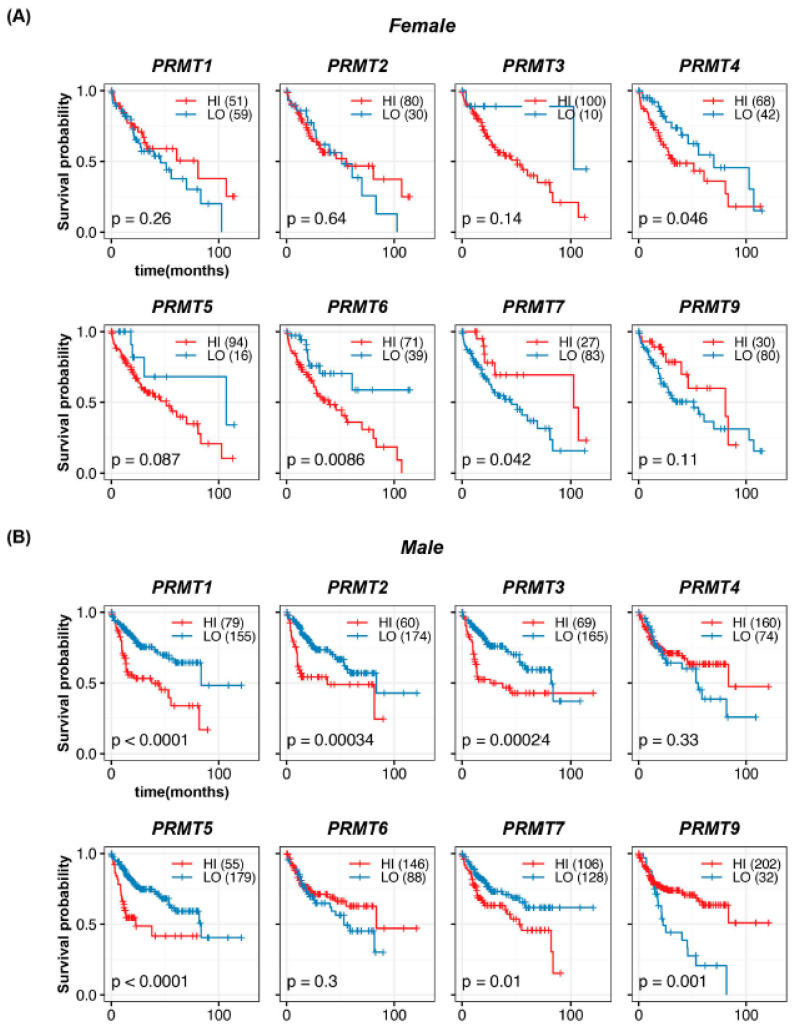
Gender-specific correlations of PRMTs in human HCC**.** (**A**,**B**) Survival probabilities (Kaplan–Meier curves) in the female (**A**) and male (**B**) patients with HCC show the gender-specific association between hepatic PRMT expression and overall survival. The R package, *multipleROC*, divided into two groups (i.e., high vs low PRMT expression) according to the optimal gene expression level of each PRMT. The R packages, *survminer* and *survival*, estimated the survival probability.

**Figure 6 cancers-15-04183-f006:**
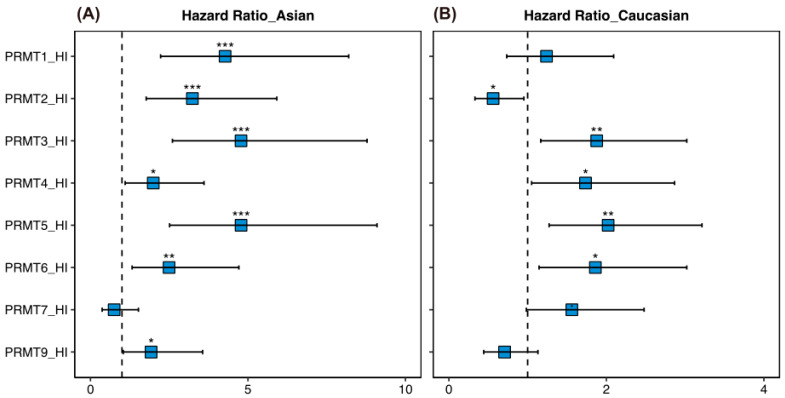
Ethnicity-specific correlations of PRMTs in human HCC. (**A**,**B**) Forest plot showing the hazard ratio and statistical significance for overall survival in Asian (**A**) and Caucasian cohorts (**B**), based on high versus low expression of PRMTs in HCC. The comparison indicated the race-specific association between hepatic PRMT expression and overall survival. The R package, *multipleROC*, divided into two groups (i.e., high vs. low PRMT expression) according to the optimal gene expression level of each PRMT. ** p* < 0.05, *** p* < 0.01, **** p* < 0.001.

**Figure 7 cancers-15-04183-f007:**
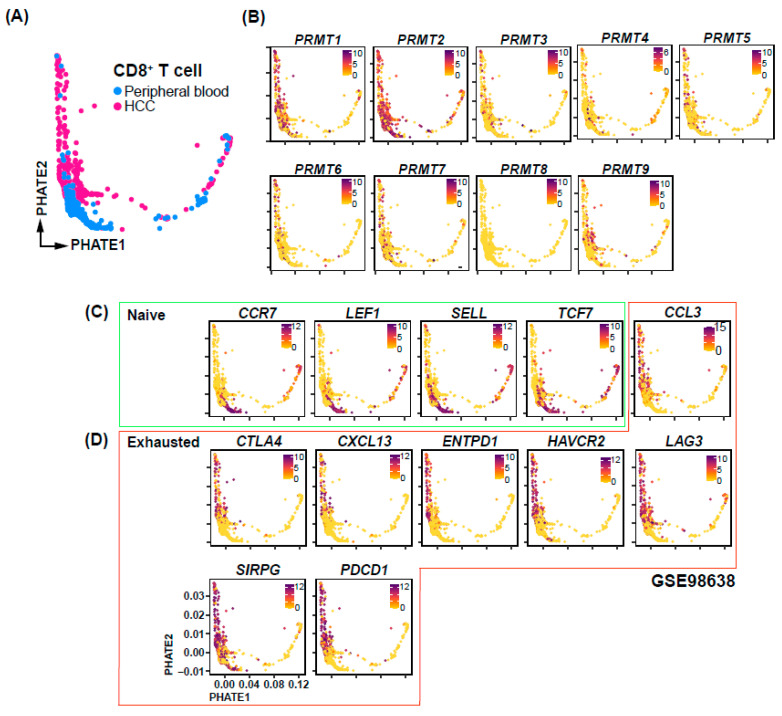
PRMT expression and immune checkpoint correlations in CD8^+^ cells of HCC patients. (**A**) UMAP-based Trajectory Analysis elucidating the developmental trajectory of CD8^+^ T cells in peripheral blood and HCC samples. (**B**) UMAP visualization identifying the differential expression patterns of PRMT family genes across distinct cell types. (**C**,**D**) Cell type-specific UMAPs representing the transcriptional profiles of genes in naïve (**C**) and exhausted (**D**) condition.

**Figure 8 cancers-15-04183-f008:**
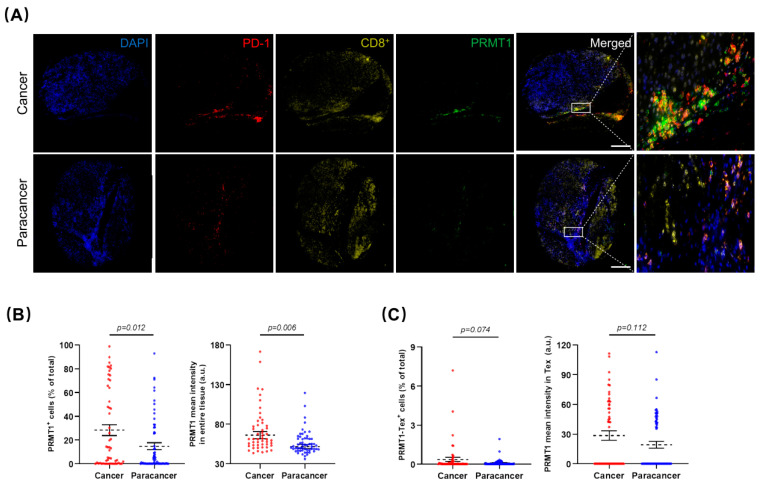
Prognostic significance of PRMT1 and PRMT1-positive T cell exhaustion (Tex) in HCC patients. (**A**) Representative multiplex immunofluorescent staining to show the expression patterns of PRMT1 (Green) in HCC and normal tissues, as well as in Tex with PD-1^+^ (Red) and CD8^+^ (Yellow) co-expression. (**B**,**C**) The expression patterns (positive rate and expression intensity) of PRMT1 (**B**) and PRMT1-positive Tex (PRMT1-Tex) (**C**) are statistically higher in HCC compared to normal tissues. (**D**,**E**) Patients are divided into two groups for further comparisons by the positive rates of PRMT1 in HCC (**D**) or in Tex of HCC (**E**), as high expression (Hi) as top 25% and low expression (Lo) as bottom 75%. (**F**) Survival probabilities (Kaplan–Meier curves) show the correlation between PRMT1 (or PRMT1-Tex) expression and overall survival. (**G**) Survival probabilities in the male patients with HCC show the association between PRMT1 (or PRMT1-Tex) expression and overall survival. (**H**,**I**) Survival probabilities in Stage I + II (**H**) and Stage III + IV (**I**) of HCC show the association between PRMT1 (or PRMT1-Tex) expression and overall survival. The R package, *multipleROC*, divided into two groups (i.e., Hi vs. Lo) according to the optimal gene expression level of PRMT1 (or PRMT1-Tex). The R packages, *survminer* and *survival*, estimated the survival probability. All results related to HCC patients and their clinical records in this figure were performed on and obtained from liver HCC microarray.

## Data Availability

Publicly available datasets were analyzed in this study. These datasets can be found in the TCGA (Accessed on 1 December 2020; https://portal.gdc.cancer.gov/) and NCBI GEO (Accessed on 24 November 2021; GEO accession no: GSE14520, GSE25097, and GSE36376) databases (https://www.ncbi.nlm.nih.gov/geo/).

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
