# Peer review of "Integrative Evaluation of the Clinical Significance Underlying Protein Arginine Methyltransferases in Hepatocellular Carcinoma"

_cancers, 2023, doi:10.3390/cancers15164183_

Round 1
Reviewer 1 Report
This manuscript has described the potential roles and clinical relevance of arginine methyltransferases (PRMTs) in HCC. They addressed main unmet needs of the field of liver biology and have analyzed the physiological values of PRMTs in various stages of HCC. The manuscript has been logically organized and written. With revision, this manuscript would be considerable for publication in Cancers.
Main comments:
1. In the case of Figures 1 and 2, the authors primarily relied on gene expression data to describe each sample or group, which appears limited. Additionally, using only one type of plot, such as boxplots, might not provide sufficient credibility to the data presentation. It is suggested that alternative plot types or different analysis approaches be considered to demonstrate the role of PRMT1, PRMT3, PRMT4, and PRMT5 as potential markers for HCC more effectively.
2. In Figure 1, the claim that PRMT2's expression is inconsistent to be a marker contrasts with its potential marker designation based on TNM stage classification in Figure 2. The connection between these findings needs clarification.
3. In Figure 3, the criteria for classifying high- and low-expression groups could be better demonstrated using a violin plot or other suitable data representation.
4. The authors categorized the stages differently in Figure 2 (1, 2, and 3+4) and Figure 4 (1+2 and 3+4) when showing PRMTs' expression in HCC patients and survival rates, respectively. An explanation for this discrepancy would be beneficial.
5. The use of only Kaplan-Meier plots for survival rates may lead to a monotonous presentation. Including additional data and utilizing diverse data presentation methods could enhance the analysis.
6. The authors' intentions and claims regarding PRMTs' expression assessment using the same data but different criteria need to be clarified to address the ambiguity in the interpretation.
Minor comments:
1. What is the rationale behind grouping TNM stages 3 and 4 together in line 256? Is it due to the limited sample size for each stage or because distinguishing them becomes clinically irrelevant?
2. In line 281, despite PRMT6 not showing any significant differences earlier, it has now emerged as an influential factor in survival. What could be the underlying reason for this observed impact?
3. In line 288, what criteria were used to combine TNM stages 1+2 and stages 3+4? Alternatively, does the literature provide any references or guidelines for this specific grouping?
4. In line 328, can we attribute the observed differences solely to racial factors? It is interesting whether other factors, such as environmental influences, were taken into consideration.
5. In line 379, the comparison was made between Asian and Caucasian populations, but only Asians were analyzed. Is this due to the higher incidence rate among Asians, as mentioned in line 328?
6. Although line 512 specifies a focus on China, line 62 mentions Mongolia and Thailand as Asian countries, and line 484 includes Mongolia, Thailand, Cambodia, and Vietnam. Why was China chosen specifically, and how representative are Chinese patients in reflecting the broader Asian population in this study?
Author Response
Dear Respected Reviewer, please see the attachment in the word file. We would like to express our sincere gratitude for your comments!!!

Reviewer 2 Report
Overall, the paper titled "Integrative Evaluation of the Clinical Significance Underlying Protein Arginine Methyltransferases in Hepatocellular Carcinoma" presents an interesting and timely investigation into the role of protein arginine methyltransferases (PRMTs) in hepatocellular carcinoma (HCC). The study combines multiple experimental approaches and clinical data analysis to provide valuable insights into the potential clinical significance of PRMTs in HCC. The work demonstrates the authors' commitment to advancing our understanding of this complex and clinically relevant field. However, several major and minor concerns need to be addressed before considering publication:
Major Concerns:
-
1.Lack of clarity in the objectives: The introduction section should be more focused on outlining the specific research questions and objectives of the study. The authors need to clearly state the hypotheses they aimed to test and the rationale behind their choice of PRMTs for investigation in HCC. Why does the author want to highlight PRMT1? According to the author's conclusion, it is difficult to understand the logic behind this. The article is all-encompassing but lacks logic.
-
2. Sample size and statistical power: The authors should provide a detailed explanation of the sample size calculation and justify the selected sample size used in both the experimental and clinical analysis. Additionally, a power analysis should be performed to ensure that the study has sufficient statistical power to detect the reported effect sizes.
- 3. Validation of findings: While the findings are intriguing, there is a lack of external validation of the results obtained. The authors should validate their findings using an independent dataset or in vitro/in vivo experiments to confirm the observed associations between PRMTs and HCC.
4. Interpretation of results: The authors need to be cautious in their interpretations and should avoid overgeneralizing the results. A more nuanced discussion of the observed correlations and potential underlying mechanisms is warranted.
Author Response

(The authors gave the same response as above.)

Reviewer 3 Report
This study identifies protein arginine methyltransferases (PRMTs), specifically PRMT1, as potential molecular targets in hepatocellular carcinoma (HCC). By analyzing multiple HCC cohorts, the expression level of PRMT1 was significantly associated with HCC progression and prognosis. Overexpression of PRMT1, particularly in males, correlated with poor prognosis and heightened HCC pathological progression. Intriguingly, PRMT1 was detected in a subset of CD8+ T cells linked with T cell exhaustion (Tex), justifying the poor prognosis of high PRMT1 levels in HCC. More significant percentages of PRMT1-Tex+ cells were found in patients with PRMT1 overexpression, strongly correlating with mortality rates, suggesting PRMT1 as a Tex status indicator. HCC was more common in males, with PRMT1 particularly impacting male prognosis. Overexpressed PRMTs, specifically PRMT1, are related to poor prognosis in Asian HCC patients, potentially due to genetic and lifestyle factors.
Overall impression of this manuscript is good, although it lacks novelty in the cancer field.
Major point
While the authors provide valuable insights into the clinical significance of protein arginine methyltransferases (PRMTs) in hepatocellular carcinoma (HCC), the study was conducted using a relatively small sample size of 344 HCC patients from the TCGA dataset, which may limit the generalizability of the findings. Additionally, the study did not include a control group of healthy individuals, which makes it difficult to determine whether the observed changes in PRMT expression are specific to HCC or a more general response to cancer. Using datasets besides the TCGA dataset should also be considered to strengthen their findings further.
Minor point
In Figure 1, the graph's Y-axis is not unified, and the microarray data is graphed with arbitrary numerical values. The Y-axis should be unified to values from 0. Additionally, it would be easier for the reader to understand if adding a note to the legend in Fig 1 that the data are microarray results.
Figure 8A shows essential data indicating the relationship between PRMT1 and Tex, but the picture is too small to tell. At least an enlarged view of the PRMT1-positive areas should be shown, including the Merge diagram.
Author Response

(The authors gave the same response as above.)

Round 2
Reviewer 1 Report
The authors did not spare any efforts to revise the manuscript.
The revised one is now appropriate for the publication.
Reviewer 2 Report
The revised manuscript improved greatly and the author answers well all the comments. Agree to accept this version.
Reviewer 3 Report
The authors very carefully made revisions in response to comments from reviewers. My concerns about the original manuscript have been swept away, and the revised manuscript has improved in quality. Therefore I am happy to accept it.